# Synthesizing Programmatic Policies that Inductively Generalize

**Jeevana Priya Inala**
MIT CSAIL
jinala@csail.mit.edu

**Osbert Bastani**
University of Pennsylvania
obastani@seas.upenn.edu

**Zenna Tavares**
MIT CSAIL
zenna@mit.edu

**Armando Solar-Lezama**
MIT CSAIL
asolar@csail.mit.edu

## Abstract

Deep reinforcement learning has successfully solved a number of challenging control tasks. However, learned policies typically have difficulty generalizing to novel environments. We propose an algorithm for learning *programmatic state machine policies* that can capture repeating behaviors. By doing so, they have the ability to generalize to instances requiring an arbitrary number of repetitions, a property we call *inductive generalization*. However, state machine policies are hard to learn since they consist of a combination of continuous and discrete structures. We propose a learning framework called *adaptive teaching*, which learns a state machine policy by imitating a teacher; in contrast to traditional imitation learning, our teacher adaptively updates itself based on the structure of the student. We show that our algorithm can be used to learn policies that inductively generalize to novel environments, whereas traditional neural network policies fail to do so.

## 1 Introduction

Existing deep reinforcement learning (RL) approaches have difficulty generalizing to novel environments (Packer et al., 2018; Rajeswaran et al., 2017). More specifically, consider a task that requires performing a repeating behavior—we would like to be able to learn a policy that generalizes to instances requiring an arbitrary number of repetitions. We refer to this property as *inductive generalization*. In supervised learning, specialized neural network architectures have been proposed that exhibit inductive generalization on tasks such as list manipulation (Cai et al., 2017), but it is not obvious how those techniques would generalize to the control problems discussed in this paper. Alternatively, algorithms have been proposed for learning programmatic policies for control problems that generalize better than traditional neural network policies (Verma et al., 2019; 2018), but existing approaches have focused on simple stateless policies that make learning generalizable repetitive behaviors hard, e.g., a stateless program cannot internally keep track of the number of repetitions made so far and decide the next action based on that progress.

We propose an algorithm for learning *programmatic state machine policies*. Such a policy consists of a set of internal states, called *modes*, each of which is associated with a controller that is applied while in that mode. The policy also includes transition predicates that describe how the mode is updated. These policies are sufficiently expressive to capture tasks of interest—e.g., they can perform repeating tasks by cycling through some subset of modes during execution. Additionally, state machine policies are strongly biased towards policies that inductively generalize, that deep RL policies lack. In other words, this policy class is both *realizable* (i.e., it contains a "right" policy that solves the problem for all environments) and *identifiable* (i.e., we can learn the right policy from limited data).

However, state machine policies are challenging to learn because their discrete state transitions make it difficult to use gradient-based optimization. One standard solution is to "soften" the state transitions by making them probabilistic. However, these techniques alone are insufficient; they still run into local optima due to the the constraints on the structure of the policy function, as well as the relatively few parameters they possess.

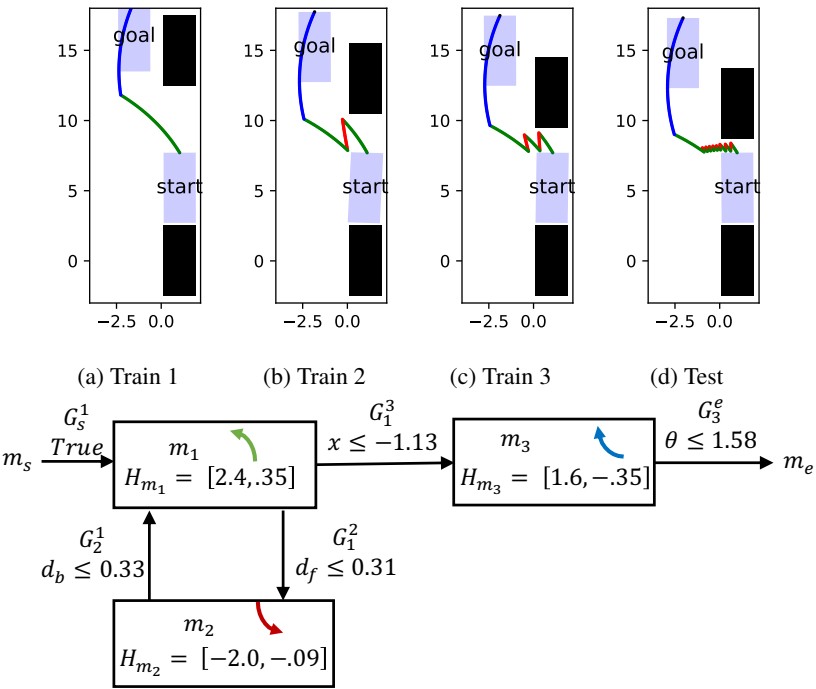

(a) Train 1      (b) Train 2      (c) Train 3      (d) Test

(e) State machine based policy. False edges are dropped.

Figure 1: Running example: retrieving an autonomous car from tight parking spots. The goal is to learn a state-machine policy (e) that is trained on scenarios (a), (b), and (c), that generalizes to scenario (d).

To address this issue, we propose an approach called *adaptive teacher* that alternatingly learns a teacher and a student. The teacher is an over-parameterized version of the student, which is a state-machine policy trained to mimic the teacher. Because the teacher is over-parameterized, it can be easily learned using model-based numerical optimization (but does not generalize as well as the student). Furthermore, our approach is different from traditional imitation learning (Schaal, 1999; Ross et al., 2011) since the teacher is regularized to favor strategies similar to the ones taken by the student, to ensure the student can successfully mimic the teacher. As the student improves, the teacher improves as well. This alternating optimization can naturally be derived within the framework of variational inference, where the teacher encodes the variational distribution (Wainwright et al., 2008).

We implement our algorithm and evaluate it on a set of reinforcement learning problems focused on tasks that require inductive generalization. We show that traditional deep RL approaches perform well on the original task, but fail to generalize inductively, whereas our state machine policies successfully generalize beyond the training distribution.

We emphasize that we do not focus on problems that require large state-machines, which is a qualitatively different problem from ours and would require different algorithms to solve. We believe that state-machines are most useful when only a few modes are required. In particular, we are interested in problems where a relatively simple behavior must be repeated a certain number of times to solve the given task. The key premise behind our approach, as shown by our evaluation, is that, in these cases, compact state-machines can represent policies that both have good performance and are generalizable. In fact, our algorithm solves all of our benchmarks using state-machine policies with at most 4 modes. When many modes are needed, then the number of possible transition structures grows exponentially, making it difficult to learn the "right" structure without having an exponential amount of training data.

**Example.** Consider the autonomous car in Figure 1, which consists of a blue car (the agent) parked between two stationary black cars. The system state is $(x, y, \theta, d)$, where $(x, y)$ is the center of the car, $\theta$ is the orientation, and $d$ is the distance between the two black cars. The actions are $(v, \psi)$,

where $v$ is velocity and $\psi$ is steering angle (we consider velocity control since the speed is low). The dynamics are standard bicycle dynamics. The goal is to drive out of the parked spot to an adjacent lane while avoiding collisions. This task is easy when $d$ is large (Figure 1a). It is somewhat more involved when $d$ is small, since it requires multiple maneuvers (Figures 1b and 1c). However, it becomes challenging when $d$ is very small (Figure 1d). A standard RL algorithm will train a policy that performs well on the distances seen during training but does not generalize to smaller distances. In contrast, our goal is to train an agent on scenarios (a), (b), and (c), that generalizes to scenario (d).

In Figure 1e, we show a state machine policy synthesized by our algorithm for this task. We use $d_f$ and $d_b$ to denote the distances between the agent and the front and back black cars, respectively. This policy has three different modes (besides a start mode $m_s$ and an end mode $m_e$). Roughly speaking, this policy says (i) immediately shift from mode $m_s$ to $m_1$, and drive the car forward and to the left, (ii) continue until close to the car in front; then, transition to mode $m_2$, and drive the car backwards and to the right, (iii) continue until close to the car behind; then, transition back to mode $m_1$, (iv) iterate between $m_1$ and $m_2$ until the car can safely exit the parking spot; then, transition to mode $m_3$, and drive forward and to the right to make the car parallel to the lane. This policy inductively generalizes since it captures the iterative behavior of driving forward and then backward until exiting the parking spot. Thus, it successfully solves the scenario in Figure 1d.

**Related work.** There has been growing interest in using program synthesis to aid machine learning (Lake et al., 2015; Ellis et al., 2015; 2018; Valkov et al., 2018; Young et al., 2019). Our work is most closely related to recent work using imitation learning to learn programmatic policies (Verma et al., 2018; Bastani et al., 2018; Zhu et al., 2019; Verma et al., 2019). These approaches use a neural network policy as the teacher. However, they are focused on learning stateless policies and hence, they use a supervised dataset of state-action pairs from the teacher and a domain-specific program synthesizer to learn programmatic policies. Building such a synthesizer for state machine policies is challenging since they contain both discrete and continuous parameters and internal state. The student in our algorithm needs to learn the state-machine policy from entire "trajectory traces" to learn the internal state. In particular, each trajectory trace consists of the sequence of states and actions from the initial state to the goal state visited by the teacher, but also encodes which states correspond to mode changes for the teacher. In the teacher's iteration, the teacher's mode changes are regularized to align more closely with the possible student mode changes. As a consequence, in the student's iteration, it is easier for the student to mimic the teacher's mode changes. Leveraging this connection between the teacher structure and student structure is critical for us to be able to learn state-machine policies Additionally, with the exception of (Verma et al., 2019), for the other approaches, there is no feedback from the student to the teacher.

State machines have been previously used to represent policies that have internal state (typically called *memory*). To learn these policies, gradient ascent methods assume a fixed structure and optimize over real-valued parameters (Meuleau et al., 1999; Peshkin et al., 2001; Aberdeen & Baxter, 2002), whereas policy iteration methods uses dynamic programming to extend the structure (Hansen, 1998). Our method combines both, but similarly to Poupart & Boutilier (2004), the structure space is bounded. In addition, programmatic state machines use programs to represent state transitions and actions rules, and as a result can perform well while remaining small in size. Hierarchies of Abstract Machines (HAM)s also use programmatic state machines for hierarchical reinforcement learning, but assumed a fixed, hand-designed structure (Parr & Russell, 1998; Andre & Russell, 2002).

Our inductive generalization goal is related to that of meta-learning (Finn et al., 2017); however, whereas meta-learning trains on a few examples from the novel environment, our goal is to generalize without additional training. Our work is also related to guided policy search, which uses a teacher in the form of a trajectory optimizer to train a neural network student (Levine & Koltun, 2013). However, training programmatic policies is more challenging since the teacher must mirror the structure of the student. Finally, it has recently been shown that over-parameterization is essential in helping neural networks avoid local minima (Allen-Zhu et al., 2019). Relaxing optimization problems by adding more parameters is a well established technique; in many cases, re-parameterization can make difficult non-convex problems solve efficiently (Carlone & Calafiore, 2018).

## 2 PROBLEM FORMULATION

**Dynamics.** We are interested in synthesizing control policies for deterministic, continuous-time dynamical systems with continuous state and action spaces. In particular, we consider partially observable Markov decision processes (POMDP) $\langle \mathcal{X}, \mathcal{A}, \mathcal{O}, F, Z, X_0, \phi_S, \phi_G \rangle$ with states $\mathcal{X} \subseteq \mathbb{R}^{d_X}$, actions $\mathcal{A} \subseteq \mathbb{R}^{d_A}$, observations $\mathcal{O} \subseteq \mathbb{R}^{d_O}$, deterministic dynamics $F : \mathcal{X} \times \mathcal{A} \to \mathcal{X}$ (i.e., $\dot{\mathbf{x}} = F(\mathbf{x}, \mathbf{a})$), deterministic observation function $Z : \mathcal{X} \to \mathcal{O}$, and initial state distribution $\mathbf{x}_0 \sim X_0$.

We consider a safety specification $\phi_S : \mathcal{X} \to \mathbb{R}$ and a goal specification $\phi_G : \mathcal{X} \to \mathbb{R}$. Then, the agent aims to reach a goal state $\phi_G(\mathbf{x}) \leq 0$ while staying in safe states $\phi_S(\mathbf{x}) \leq 0$. A positive value for $\phi_S(\mathbf{x})$ (resp., $\phi_G(\mathbf{x})$) quantifies the degree to which $\mathbf{x}$ is unsafe (resp., away from the goal).

**Policies.** We consider policies with internal memory $\pi : \mathcal{O} \times \mathcal{S} \to \mathcal{A} \times \mathcal{S}$ where $\mathcal{S} \subseteq \mathbb{R}^{d_S}$ is the set of internal states; we assume the memory is initialized to a constant $\mathbf{s}_0$. Given such a policy $\pi$, we sample a rollout (or trajectory) $\tau = (\mathbf{x}_0, \mathbf{x}_1, ..., \mathbf{x}_N)$ with horizon $N \in \mathbb{N}$ by sampling $\mathbf{x}_0 \sim X_0$ and then performing a discrete-time simulation $\mathbf{x}_{n+1} = \mathbf{x}_n + F(\mathbf{x}_n, \mathbf{a}_n) \cdot \Delta$, where $(\mathbf{a}_n, \mathbf{s}_{n+1}) = \pi(Z(\mathbf{x}_n), \mathbf{s}_n)$ and $\Delta \in \mathbb{R}_{>0}$ is the time increment. Since $F$, $Z$, and $\pi$ are deterministic, $\tau$ is fully determined by $\mathbf{x}_0$ and $\pi$; $\tau$ can also be represented as a list of actions combined with the initial state i.e $\tau = \langle \mathbf{x}_0, (\mathbf{a}_0, \mathbf{a}_1, \cdots, \mathbf{a}_N) \rangle$.

The degree to which $\phi_S$ and $\phi_G$ are satisfied along a trajectory is quantified by a reward function $R(\pi, \mathbf{x}_0) = -\phi_G(\mathbf{x}_N)^+ - \sum_{n=0}^{N} \phi_S(\mathbf{x}_n)^+$, where $x^+ = \max(0, x)$. The optimal policy $\pi^*$ in some policy class $\Pi$ is one which maximizes the expected reward $\mathbb{E}_{\mathbf{x}_0 \sim X_0}[R(\pi, \mathbf{x}_0)]$.

**Inductive generalization.** Beyond optimizing reward, we want a policy that *inductively generalizes* to unseen environments. Formally, we consider two initial state distributions: a training distribution $X_0^{\text{train}}$, and a test distribution $X_0^{\text{test}}$ that includes the extreme states never encountered during training. Then, the goal is to train a policy according to $X_0^{\text{train}}$—i.e.,

$$\pi^* = \arg\max_{\pi \in \Pi} \mathbb{E}_{\mathbf{x}_0 \sim X_0^{\text{train}}}[R(\pi, \mathbf{x}_0)], \tag{1}$$

but measure its performance according to $X_0^{\text{test}}$—i.e., $\mathbb{E}_{\mathbf{x}_0 \sim X_0^{\text{test}}}[R(\pi, \mathbf{x}_0)]$.

## 3 PROGRAMMATIC STATE MACHINE POLICIES

To achieve inductive generalization, we aim to synthesize programmatic policies in the form of state machines. At a high level, state machines can be thought of as compositions of much simpler policies, where the internal state of the state machines (called its mode) indicates which simple policy is currently being used. Thus, state machines are capable of encoding complex nonlinear control tasks such as iteratively repeating a complex sequence of actions (e.g., the car example in Figure 1). At the same time, state machines are substantially more structured than more typical policy classes such as neural networks and decision trees.

More precisely, a state machine $\pi$ is a tuple $\langle \mathcal{M}, \mathcal{H}, \mathcal{G}, m_s, m_e \rangle$. The *modes* $m_i \in \mathcal{M}$ of $\pi$ are the internal memory of the state machine. Each mode $m_i \in \mathcal{M}$ corresponds to an *action function* $H_{m_i} \in \mathcal{H}$, which is a function $H_{m_i} : \mathcal{O} \to \mathcal{A}$ mapping observations to actions. When in mode $m_i$, the agent takes action $\mathbf{a} = H_{m_i}(\mathbf{o})$. Furthermore, each pair of modes $(m_i, m_j)$ corresponds to a *switching condition* $G_{m_i}^{m_j} \in \mathcal{G}$, which is a function $G_{m_i}^{m_j} : \mathcal{O} \to \mathbb{R}$. When an agent in mode $m_i$ observes $\mathbf{o}$ such that $G_{m_i}^{m_j}(\mathbf{o}) \geq 0$, then the agent transitions from mode $m_i$ to mode $m_j$. If there are multiple modes $m_j$ with non-negative switching weight $G_{m_i}^{m_j}(\mathbf{o}) \geq 0$, then the agent transitions to the one that is greatest in magnitude; if there are several modes of equal weight, we take the first one according to a fixed ordering. Finally, $m_s, m_e \in \mathcal{M}$ are the start and end modes, respectively; the state machine mode is initialized to $m_s$, and the state machine terminates when it transitions to $m_e$.

Formally, $\pi(\mathbf{o}_n, \mathbf{s}_n) = (\mathbf{a}_n, \mathbf{s}_{n+1})$, where $\mathbf{a}_n = H_{\mathbf{s}_n}(\mathbf{o}_n)$, $\mathbf{s}_0 = m_s$ and

$$\mathbf{s}_{n+1} = \begin{cases} m^* = \text{darg}\max_m G_{\mathbf{s}_n}^m(\mathbf{o}_n) & \text{if } G_{\mathbf{s}_n}^{m^*}(\mathbf{o}_n) \geq 0 \\ \mathbf{s}_n & \text{otherwise} \end{cases} \tag{2}$$

where $\text{darg}\max$ is a deterministic $\arg\max$ that breaks ties as described above.

Action functions and switching conditions are specified by *grammars* that encode the space of possible functions as a space of programs. Different grammars can be used for different problems.

Typical grammars for action functions include constants $\{C_\alpha : \mathbf{o} \mapsto \alpha\}$ and proportional controls $\{P^i_{\alpha_0,\alpha_1} : \mathbf{o} \mapsto \alpha_0(\mathbf{o}[i] - \alpha_1)\}$. A typical grammar for switching conditions is the grammar

$$B ::= \{\mathbf{o}[i] \leq \alpha\}_i \mid \{\mathbf{o}[i] \geq \alpha\}_i \mid B_1 \wedge B_2 \mid B_1 \vee B_2$$

of Boolean predicates over the current observation $\mathbf{o}$, where $\mathbf{o}[i]$ is the $i$th component of $\mathbf{o}$. In all these grammars, $\alpha_i \in \mathbb{R}$ are parameters to be learned. The grammar for switching conditions also has discrete parameters encoding the choice of expression. For example, in Figure 1, the action functions are constants, and the switching conditions are inequalities over components of $\mathbf{o}$.

## 4 FRAMEWORK FOR SYNTHESIZING PROGRAMMATIC POLICIES

We now describe the adaptive teaching framework for synthesizing state machine policies. In this section, the teacher is abstractly represented as a collection of trajectories $\tau_{\mathbf{x}_0}$ (i.e., an open-loop controller consisting of a fixed sequence of actions) for each initial state $\mathbf{x}_0$. A key insight is that we can parameterize $\tau_{\mathbf{x}_0}$ in a way that mirrors the structure of the state machine student. As we discuss in Section 4.2, we parameterize $\tau_{\mathbf{x}_0}$ as a "loop-free" state machine. Intuitively, our algorithm efficiently computes $\tau_{\mathbf{x}_0}$ (from multiple initial states $\mathbf{x}_0$) using gradient-based optimization, and then "glues" them together using maximum likelihood to construct a state machine policy.

### 4.1 ADAPTIVE TEACHING VIA VARIATIONAL INFERENCE

We derive the adaptive teaching formulation by reformulating the learning problem in the framework of probabilistic reinforcement learning, and also consider policies $\pi$ that are probabilistic state machines (see Section 4.3). Then, we use a variational approach to break the problem into the teacher and the student steps. In this approach, the log-likelihood of a policy $\pi$ is defined as follows:

$$\ell(\pi) = \log \mathbb{E}_{p(\tau|\pi)}[e^{\lambda R(\tau)}] \tag{3}$$

where $p(\tau \mid \pi)$ is the probability of sampling rollout $\tau$ when using policy $\pi$ from a random initial state $\mathbf{x}_0$, $\lambda \in \mathbb{R}_{\geq 0}$ is a hyperparameter, and $R(\tau)$ is the reward assigned to $\tau$. We have

$$\ell(\pi) = \log \mathbb{E}_{q(\tau)} \left[ e^{\lambda R(\tau)} \cdot \frac{p(\tau \mid \pi)}{q(\tau)} \right] \geq \mathbb{E}_{q(\tau)}[\lambda R(\tau) + \log p(\tau|\pi) - \log q(\tau)] \tag{4}$$

where $q(\tau)$ is the variational distribution and the inequality follows from Jensen's inequality. Thus, we can optimize $\pi$ by maximizing the lower bound Eq (4) on $\ell(\pi)$. Since the first and third term of Eq (4) are constant with respect to $\pi$, we have

$$\pi^* = \arg\max_\pi \mathbb{E}_{q(\tau)}[\log p(\tau|\pi)]. \tag{5}$$

Next, the optimal choice for $q$ (i.e., to minimize the gap in the inequality in Eq (4)) is

$$q^* = \arg\min_q D_{\mathrm{KL}}(q(\tau) \parallel e^{\lambda R(\tau)} \cdot p(\tau \mid \pi)/Z) \tag{6}$$

where $Z$ is a normalizing constant. We choose $q$ to have form $q(\tau) = p(\mathbf{x}_0) \cdot \delta(\tau - \tau_{\mathbf{x}_0})$, where $\delta$ is the Dirac delta function, $p(\mathbf{x}_0)$ is the initial state distribution, and $\tau_{\mathbf{x}_0}$ are the parameters to be optimized, where $\tau_{\mathbf{x}_0}$ encodes a trajectory from $\mathbf{x}_0$. Then, up to constants, the objective of Eq (6) equals

$$\mathbb{E}_{p(\mathbf{x}_0)} \left[ \log p(\mathbf{x}_0) + \mathbb{E}_{\delta(\tau-\tau_{\mathbf{x}_0})}[\log \delta(\tau - \tau_{\mathbf{x}_0})] - (\lambda R(\tau_{\mathbf{x}_0}) + \log p(\tau_{\mathbf{x}_0} \mid \pi, \mathbf{x}_0)) \right].$$

The first term is constant; the second term is degenerate, but it is also constant. Thus, we have

$$q^* = \arg\max_{\{\tau_{\mathbf{x}_0}\}} \mathbb{E}_{p(\mathbf{x}_0)} \left[ \lambda R(\tau_{\mathbf{x}_0}) + \log p(\tau_{\mathbf{x}_0} \mid \pi, \mathbf{x}_0) \right]. \tag{7}$$

Thus, we can optimize Eq (3) by alternatingly optimizing Eq (5) and Eq (7).

We interpret these equations as adaptive teaching. At a high level, the teacher (i.e., the variational distribution $q^*$ in Eq (7)) is used to guide the optimization of the student (i.e., the state machine policy $\pi^*$ in Eq (5)). Rather than compute the teacher in closed form, we approximate it by sampling

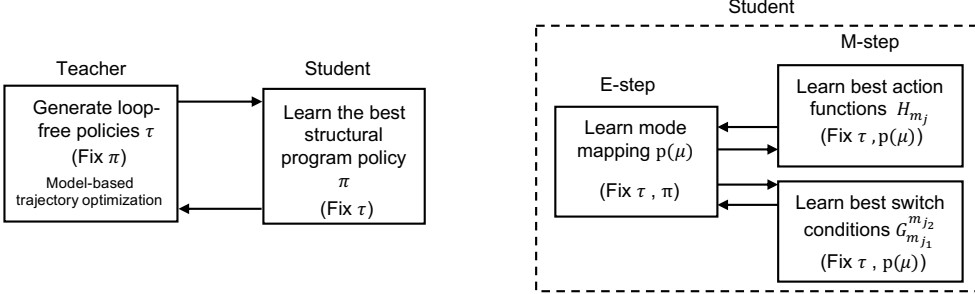

Figure 2: Flowchart connecting the different components of the algorithm.

finitely many initial states $\mathbf{x}_0^k \sim X_0$ and then computing the optimal rollout from $\mathbf{x}_0^k$. Formally, on the $i$th iteration, the teacher and student are updated as follows:

$$\textbf{Teacher} \quad q_i^* = \sum_{k=1}^{K} \delta(\tau_k^i) \tag{8}$$

$$\text{where} \quad \tau_k^i = \arg\max_{\tau} \ \lambda R(\tau) + \log p(\tau \mid \pi^{i-1}, \mathbf{x}_0^k) \quad (\mathbf{x}_0^k \sim X_0)$$

$$\textbf{Student} \quad \pi_i^* = \arg\max_{\pi} \ \sum_{k=1}^{K} \log p(\tau_k^i \mid \pi, \mathbf{x}_0^k) \tag{9}$$

The teacher objective Eq (8) is to both maximize the reward $R(\tau)$ from a random initial state $\mathbf{x}_0$ and to maximize the probability $p(\tau \mid \pi, \mathbf{x}_0)$ of obtaining the rollout $\tau$ from initial state $\mathbf{x}_0$ according to the current student $\pi$. The latter encourages the teacher to match the structure of the student. Furthermore, the teacher is itself updated at each step to account for the changing structure of the student. The student objective Eq (9) is to imitate the distribution of rollouts according to the teacher. Figure 2 shows the different components of our algorithm.

## 4.2 Teacher: Computing Loop-Free Policies

We begin by describing how the teacher solves the trajectory optimization problem Eq (8)—i.e., computing $\tau_k$ for a given initial state $\mathbf{x}_0^k$.

**Parameterization.** One approach is to parameterize $\tau$ as an arbitrary action sequence $(\mathbf{a}_0, \mathbf{a}_1, ...)$ and use gradient-based optimization to compute $\tau$. However, this approach can perform poorly—even though we regularize $\tau$ towards the student, it could exhibit behaviors that are hard for the student to capture. Instead, we parameterize $\tau$ in a way that mirrors the student. In particular, we parameterize $\tau$ like a state machine, but rather than having modes and switching conditions that adaptively determine the sequence of action functions to be executed and the duration of execution, the sequence of action functions is fixed and each action function is executed for a fixed duration.

More precisely, we represent $\tau$ as an *loop-free policy* $\tau = \langle \mathcal{H}, \mathcal{T} \rangle$. To execute $\tau$, each action function $H_i \in \mathcal{H}$ is applied for the corresponding duration $T_i \in \mathcal{T}$, after which $H_{i+1}$ is applied. The action functions are from the same grammar of action functions for the student.

The obvious way to represent a duration $T_i$ is as a number of time steps $T_i \in \mathbb{N}$. However, with this choice, we cannot use continuous optimization to optimize $T_i$. Instead, we fix the number of discretization steps $P$ for which $H_i$ is executed, and vary the time increment $\Delta_i = T_i/P$—i.e., $\mathbf{x}_{n+1} \approx \mathbf{x}_n + F(\mathbf{x}_n, H_i(\mathbf{o})) \cdot \Delta_i$. We enforce $\Delta_i \leq \Delta_{\max}$ for a small $\Delta_{\max}$ to ensure that the discrete-time approximation of the dynamics is sufficiently accurate.

Figure 3(a) and (d) show examples of loop-free policies for two different initial states and two different teacher iterations. The loop-free policies in (d) are regularized to match the student's state-machine policy learned in the previous iteration (shown in Figure 3(c)).

**Optimization.** We use model-based trajectory optimization to compute loop-free policies. The main challenge is handling the term $p(\tau \mid \pi, \mathbf{x}_0)$ in the objective. Symbolically computing the

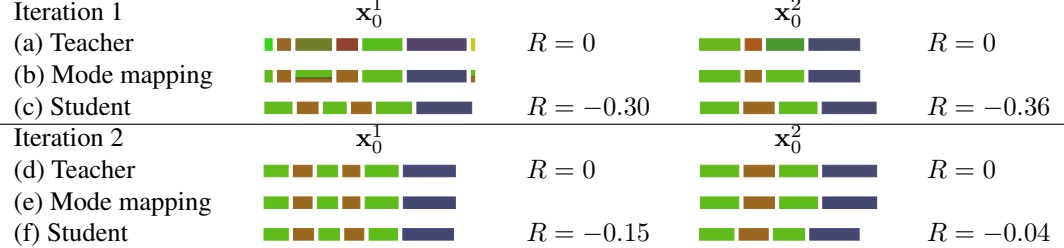

Figure 3: Visualization showing the student-teacher interaction for two iterations. (a) The loop-free policies (with their corresponding rewards) learned by the teacher for two different initial states. Here, the boxes signify the different segments in the loop-free policies, the colors signify different actions, and the lengths of the boxes signify the durations of the segments. (b) The mapping between the segments and the modes in the state-machine—i.e., $p(\mu = m_j)$. Each box shows the composition of modes vertically distributed according their probabilities. For example, the third segment in the loop-free policy for $\mathbf{x}_0^1$ has $p(\mu = \text{Green}) = 0.65$ and $p(\mu = \text{Brown}) = 0.35$. (c) The most probable rollouts from the state-machine policy learned by the student. Finally, (d), (e) and (f) are similar to (a), (b) and (c), but for the second iteration.

this probability is hard because of the discrete-continuous structure of $\pi$. Another alternative is to precompute the probabilities of all the trajectories $\tau$ that can be derived from $\pi$. However, this is also infeasible because the number of trajectories is unbounded. Thus, we perform trajectory optimization in two phases. First, we use a sampling-based optimization algorithm to obtain a set of good trajectories $\tau^1, ..., \tau^L$. Then, we apply gradient-based optimization, replacing $p(\cdot \mid \pi, \mathbf{x}_0)$ with a term that regularizes $\tau$ to be close to $\{\tau^\ell\}_{\ell=1}^L$.

The first phase proceeds as follows: (i) sample $\tau^1, \cdots, \tau^L$ using $\pi$ from $\mathbf{x}_0$, and let $p^\ell$ be the probability of $\tau^\ell$ according to $\pi$, (ii) sort these samples in decreasing order of objective $p^\ell \cdot e^{\lambda R(\tau^\ell)}$, and (iii) discard all but the top $\rho$ samples. This phase essentially performs one iteration of CEM (Mannor et al., 2003). Then, in the second phase, we replace the probability expression with $p(\tau \mid \pi, \mathbf{x}_0) \approx \frac{\sum_{\ell=1}^\rho p^\ell \cdot e^{-d(\tau, \tau^\ell)}}{\sum_{\ell=1}^\rho p^\ell}$, which we use gradient-based optimization to optimize. Here, $d(\tau, \tau^\ell)$ is a distance metric between two loop-free policies, defined as the $L_2$ distance between the parameters of $\tau$ and $\tau^\ell$. We chose the number of samples, $\rho = 10$. For our benchmarks, we did not notice any improvement in the number of student-teacher iterations by increasing $\rho$ above 10. So, we believe we are not losing any information from this approximation.

## 4.3 STUDENT: LEARNING STRUCTURED STATE MACHINE POLICIES VIA IMITATION

Next, we describe how the student solves the maximum likelihood problem Eq (9) to compute $\pi^*$.

**Probabilistic state machines.** Although the output of our algorithm is a student policy that is a deterministic state machine, our algorithm internally relies on distributions over rollouts induced by the student policy to guide the teacher. Thus, we represent the student policy as a probabilistic state machine during learning. To do so, we simply make the action functions $H_{m_j}$ and switching conditions $G_{m_{j_1}}^{m_{j_2}}$ probabilistic—instead of constant parameters in the grammar for action functions and switching conditions, now we have Gaussian distributions $\mathcal{N}(\alpha, \sigma)$. Then, when executing $\pi$, we obtain i.i.d. samples of the parameters $H'_{m_j} \sim H_{m_j}$ and $\{(G_{m_j}^{m'_j})' \sim G_{m_j}^{m'_j}\}_{m'_j}$ every time we switch to mode $m_j$, and act according to $H'_{m_j}$ and $\{(G_{m_j}^{m'_j})'\}$ until the mode switches again. By re-sampling these parameters on every mode switch, we avoid dependencies across different parts of a rollout or different rollouts. On the other hand, by not re-sampling these parameters within a mode switch, we ensure that the structure of $\pi$ remains intact within a mode.

**Optimization.** Each $\tau_k$ can be decomposed into segments $(k, i)$ where action function $H_{k,i}$ is executed for duration $T_{k,i}$. For example, each block in Figure 3(a) is a segment. Furthermore, for the student $\pi$, let $H_{m_j}$ be the action function distribution for mode $m_j$ and $G_{m_{j_1}}^{m_{j_2}}$ be the switching

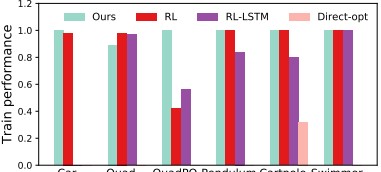 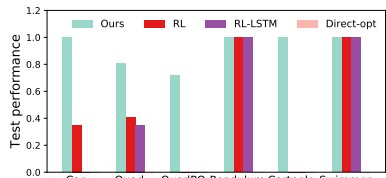 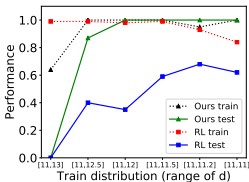

Figure 4: Comparison of performances on train (left) and test (middle) distributions. Our approach outperforms the baselines on all benchmarks in terms of test performance. An empty bar indicates that the policy learned for that experiment failed on all runs. We also plot test performance for different choices of training distribution for the Car benchmark (right).

condition distribution for mode $m_{j_1}$ to mode $m_{j_2}$. Note that $H_{m_j}$ and $G_{m_{j_1}}^{m_{j_2}}$ are distributions whereas $H_{k,i}$ and $T_{k,i}$ are constants. We have

$$p(\tau_k \mid \pi, \mathbf{x}_0^k) = \prod_i p(H_{k,i} \mid \pi, \mathbf{x}_0^k) \cdot p(T_{k,i} \mid \pi, \mathbf{x}_0^k).$$

For each $(k, i)$, let $\mu_{k,i}$ be the latent random variable indicating the $i$th mode used by $\pi$ starting from $\mathbf{x}_0^k$; in particular, $\mu_{k,i}$ is a categorical random variable that takes values in the modes $\{m_j\}$. And $\mu_{k,i} = m_j$ means that $H_{k,i}$ is sampled from the distribution $H_{m_j}$ and $T_{k,i}$ is determined by the sampled switching conditions from distributions $\{G_{m_j}^{m_j'}\}$. Assuming the latent variable $\mu_{k,i}$ allows the student to compute $\pi^*$ by computing $H_{m_j}^*$ and $G_{m_{j_1}}^{m_{j_2}*}$ separately. In Figure 3, (b) and (e) show the learned mode mappings $p(\mu = m_j)$ for the segments in the loop-free policies shown in (a) and (d) respectively.

Since directly optimizing the maximum likelihood $\pi$ is hard in the presence of the latent variables $\mu_{k,i}$, we use the standard expectation maximization (EM) approach to optimizing $\pi$, where the E-step computes the distributions $p(\mu_{k,i} = m_j)$ assuming $\pi$ is fixed, and the M-step optimizes $\pi$ assuming the probabilities $p(\mu_{k,i} = m_j)$ are fixed. See Appendix A for details. In Figure 3, (c) and (f) show the most probable rollouts from the state-machine policies learned at the end of the EM approach for two different student iterations.

## 5 EXPERIMENTS

**Benchmarks.** We use 6 control problems, each with different training and test distributions (summarized in Figure 8 in Appendix C): (i) Car, the benchmark in Figure 1, (ii) Quad, where the goal is to maneuver a 2D quadcopter through an obstacle course by controlling its vertical acceleration, where we vary the obstacle course length, see Figure 6 leftmost, (iii) QuadPO, a variant where the obstacles are unobserved but periodic (so the agent can perform well using a repeating motion), see Figure 6 (second from left), (iv) Pendulum, where we vary the pendulum mass, (v) Cart-Pole, where we vary the time horizon and pole length, and (vi) Swimmer, where the goal is to move the swimmer forward through a viscous liquid, where we vary the length of the segments comprising the robot swimmer.

**Baselines.** We compare against: (i) RL: PPO with a feed-forward neural network policy, (ii) RL-LSTM: PPO with an LSTM, (iii) Direct-Opt: learning a state machine policy directly via numerical optimization. Hyper-parameters are chosen to maximize performance on the training distribution. More details about the baselines and the hyper-parameters can be found in Appendices B.2, B.3, & B.4. Each algorithm is trained 5 times; we choose the one that performs best on the training distribution.

Note that for the comparison to RL approaches, we use model-free algorithms, whereas, in our algorithm, the teacher uses model-based optimization. We do not compare against model-based RL approaches because (a) even model-free RL approaches achieve almost perfect performance on the training distribution (see Figure 4 left) and (b) our main goal is to compare the performance of our policies and the neural network policies on the test distribution and not the training distribution. Moreover, in case the model of the system is unknown, we can use known algorithms to infer the model from data (Ahmadi et al., 2018) and then use this learned model in our algorithm.

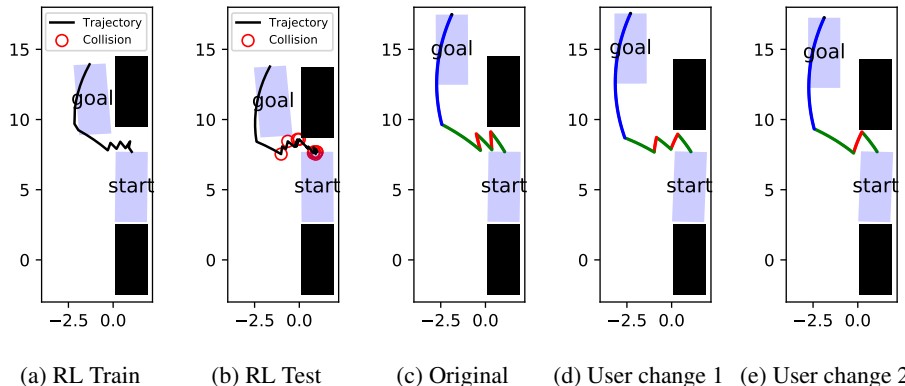

(a) RL Train  (b) RL Test  (c) Original  (d) User change 1  (e) User change 2

Figure 5: (a-c) The RL policy generates unstructured trajectories, and therefore does not generalize from (a) the training distribution to (b) the test distribution. In contrast, our state machine policy in (c) generates a highly structured trajectory that generalizes well. (c-e) A user can modify our state machine policy to improve performance. In (d), the user sets the steering angle to the maximum value 0.5, and in (e), the user sets the thresholds in the switching conditions $G_{m_1}^{m_2}, G_{m_2}^{m_1}$ to 0.1.

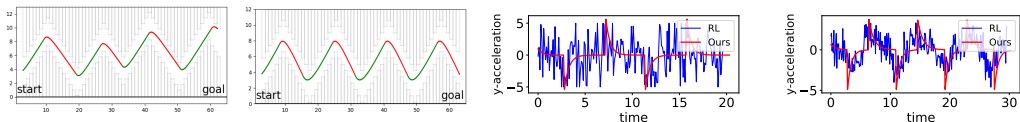

Figure 6: Left: Trajectories for the Quad (leftmost) and QuadPO (second from the left) benchmarks using our state machine policy. Right: Graph of vertical acceleration over time for both our policy (red) and the neural network policy (blue), for Quad (second from the right) and QuadPO (rightmost).

## 5.1 RESULTS

Figure 4 shows results on both training and test distributions. We measure performance as the fraction of rollouts (out of 1000) that both satisfy the safety specification and reach the goal.

**Inductive generalization.** For all benchmarks, our policy generalizes well on the test distribution. In four cases, we generalize perfectly (all runs satisfy the metric). For Quad and QuadPO, the policies result in collisions on some runs, but only towards the end of the obstacle course.

**Comparison to RL.** The RL policies mostly achieve good training performance, but generalize poorly since they over-specialize to states seen during training. The exceptions are Pendulum and Swimmer. Even in these cases, the RL policies take longer to reach the goals than our state machine policies (see Figure 10 and Figure 11 in Appendix C). For QuadPO, the RL policy does not achieve a good training performance since the states are partially observed. We may expect the LSTM policies to alleviate this issue. However, the LSTM policies often perform poorly even on the training distribution, and also generalize worse than the feed-forward neural network policies.

**Comparison to direct-opt.** The state machine policies learned using direct-opt perform poorly even in training because of the numerous local optima arising due to the structural constraints. This illustrats the need to use adaptive teaching to learn state machine policies.

## 5.2 QUALITATIVE ANALYSIS

**Behavior of policy.** We empirical analyze the policies. Figure 5 shows the trajectory taken by the RL policy (a), compared to our policy (c), from a training initial state for the Car benchmark. The RL policy does not exhibit a repeating behavior, which causes it to fail on the trajectory from a test state shown in (b). Similarly, Figure 6 (right) compares the actions taking by our policy to those taken by the RL policy on Quad and QuadPO. Our policy produces smooth repeating actions, whereas the RL policy does not. Action vs time graphs for other benchmarks can be found in the appendix (Figures 12, 13, & 14) and they all show similar behaviors.

**Varying the training distribution.** We study how the test performance changes as we vary the training distribution on the Car benchmark. We vary $X_0^{\text{train}}$ as $d \sim [d_{\min}, 13]$, where $d_{\min} = \{13, 12.5, 12, 11.5, 11.2, 11\}$, but fix $X_0^{\text{test}}$ to $d \sim [11, 12]$. Figure 4 (right) shows how test performance varies with $d_{\min}$ for both our policy and the RL policy. Our policy inductively generalizes for a wide range of training distributions. In contrast, the test performance of the RL policy initially increases as the train distribution gets bigger, but it eventually starts declining. The reason is that its training performance actually starts to decline. Thus, in some settings, our approach (even when trained on smaller distributions) can produce policies that outperform the neural network policies produced by RL (even when trained on the full distribution).

**Interpretability.** An added benefit of our state machine policies is interpretability. In particular, we demonstrate the interpretability of our policies by showing how a user can modify a learned state machine policy. Consider the policy from Figure 1e for the autonomous car. We manually make the following changes: (i) increase the steering angle in $H_{m_1}$ to its maximum value $0.5$, and (ii) decrease the gap maintained between the agent and the black cars by changing the switching condition $G_{m_1}^{m_2}$ to $d_f \leq 0.1$ and $G_{m_2}^{m_1}$ to $d_b \leq 0.1$. Figure 5 demonstrates these changes—it shows trajectories obtained using the original policy (c), the first modified policy (d), and the second modified policy (e). There is no straightforward way to make these kinds of changes to a neural network policy.

## 6 CONCLUSION

We have proposed an algorithm for learning state machine policies that inductively generalize to novel environments. Our approach is based on a framework called adaptive teaching that alternatively learns a student that imitates a teacher, who in-turn adapts to the structure of the student. We demonstrate that our policies inductively generalize better than RL policies.

In the future, we will explore more complex grammars for the action functions and the switching conditions, for example, with some parts being small neural networks, while still retaining the ability to learn generalizable behaviors. Moreover, we will extend our approach to use model-free techniques in the teacher's algorithm to make our approach more aligned with the reinforcement learning premise. Finally, we believe that the idea of learning programmatic representations and using the adaptive teaching algorithm to deal with the mixed discrete-continuous problems can be applied to other learning settings such as supervised learning and unsupervised learning.

ACKNOWLEDGMENTS

This work was supported by ONR N00014-17-1-2699 and NSF Award CCF-1910769.

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

## A  EXPECTATION MAXIMIZATION FOR STUDENT OPTIMIZATION

### A.1  COMPUTING $p(\tau \mid \pi, \mathbf{x}_0)$

First, note that we have

$$p(\tau_k \mid \pi, \mathbf{x}_0^k) = \prod_i p(H_{k,i} \mid \pi, \mathbf{x}_0^k) \cdot p(T_{k,i} \mid \pi, \mathbf{x}_0^k).$$

where

$$p(H_{k,i} \mid \pi, \mathbf{x}_0^k) = \sum_j p(H_{k,i} \mid H_{m_j}) \cdot p(\mu_{k,i} = m_j).$$

Similarly, the duration $T_{k,i}$ is determined both by the current mode $\mu_{i,k} = m_{j_1}$, and by the switching conditions $G_{m_{j_1}}^- = \{G_{m_{j_1}}^{m_{j_2}}\}_{m_{j_2}}$ from the current mode $m_{j_1}$ into some other mode $m_{j_2}$. More precisely, let $\gamma_{k,i}$ denote the trajectory (sequence of states) of the $(k,i)$ segment of $\tau_k$, and let $\zeta(\gamma_{k,i}, G_{m_j}^-)$ denote the earliest time at which a switching condition $G \in G_{m_j}^-$ becomes true along $\gamma_{k,i}$. Since $G \in G_{m_j}^-$ are distributions, $\zeta(\gamma_{k,i}, G_{m_j}^-)$ is a distribution on transition times. Then, we have

$$p(T_{k,i} \mid \pi, \mathbf{x}_0^k) = \sum_{m_{j_1}} \sum_{m_{j_2}} p(\mu_{k,i} = m_{j_1}) \cdot p(\mu_{k,i+1} = m_{j_2}) \cdot p(T_{k,i} \mid G_{m_{j_1}}^{m_{j_2}}, G_{m_{j_1}}^-)$$

$$p(T_{k,i} \mid G_{m_{j_1}}^{m_{j_2}}, G_{m_{j_1}}^-) = p(T_{k,i} = \zeta(\gamma_{k,i}, G_{m_{j_1}}^{m_{j_2}})) \cdot \prod_{m_{j_3} \neq m_{j_2}} p(T_{k,i} < \zeta(\gamma_{k,i}, G_{m_{j_1}}^{m_{j_3}})).$$

In other words, $T_{k,i}$ is the duration until $G_{m_{j_1}}^{m_{j_2}}$ triggers, conditioned on none of the conditions $G_{m_{j_1}}^{m_{j_3}}$ triggering (where $m_{j_3} \neq m_{j_2}$).

### A.2  OPTIMIZING THE STUDENT POLICY

Numerically optimizing the maximum likelihood objective to compute $\pi^*$ is hard because it requires integrating over all possible choices for the latent variables $\mu_{k,i}$. For example, if the teacher generates 10 loop-free policies every iteration and there are 10 modes in each loop-free policy, and 4 modes in the state-machine, the number of choices for the latent variables is $4^{100}$, which makes the enumeration infeasible. The expectation-maximization method provides an efficient way for computing the maximum likelihood, by alternatingly optimizing for the latent variables and the state-machine parameters. The E-step computes the probability distributions $p(\mu_{k,i} = m_j)$ for a fixed $\pi$, and the M-step optimizes $H_{m_j}$ and $G_{m_{j_1}}^{m_{j_2}}$ given $p(\mu_{k,i} = m_j)$.

**E-step.**  Assuming $\pi$ is fixed, we have

$$p(\mu_{k,i} = m_j \mid \pi, \{\tau_k\}) = \frac{p(H_{k,i} \mid H_{m_j}) \cdot p(T_{k,i} = \zeta(\gamma_{k,i}, G_{m_j}^-))}{\sum_{m_j'} p(H_{k,i} \mid H_{m_j'}) \cdot p(T_{k,i} = \zeta(\gamma_{k,i}, G_{m_j'}^-))}. \tag{10}$$

**M-step.**  Assuming $p(\mu_{k,i} = m_j)$ is fixed, we solve

$$\underset{\{H_{m_j}\}}{\arg\max} \sum_{k,i} p(\mu_{k,i} = m_j) \cdot \log p(H_{k,i} \mid H_{m_j}) \tag{11}$$

$$\underset{\{G_{m_{j_1}}^{m_{j_2}}\}}{\arg\max} \sum_{k,i} p(\mu_{k,i} = m_{j_1}) \cdot p(\mu_{k,i+1} = m_{j_2}) \cdot \log p(T_{k,i} = \zeta(\gamma_{k,i}, G_{m_{j_1}}^{m_{j_2}}))$$

$$+ p(\mu_{k,i} = m_{j_1}) \cdot (1 - p(\mu_{k,i+1} = m_{j_2})) \cdot \log p(T_{k,i} < \zeta(\gamma_{k,i}, G_{m_{j_1}}^{m_{j_2}})) \tag{12}$$

For $G_{m_{j_1}}^{m_{j_2}}$, the first term handles the case $\mu_{k,i+1} = m_{j_2}$, where we maximize the probability that $G_{m_{j_1}}^{m_{j_2}}$ makes the transition at duration $T_{k,i}$, and the second term handles the case $\mu_{k,i+1} \neq m_{j_2}$, where we maximize the probability that $G_{m_{j_1}}^{m_{j_2}}$ does not make the transition until after duration $T_{k,i}$.

We briefly discuss how to solve these equations. For action functions, suppose that $H$ encodes the distribution $\mathcal{N}(\alpha_H, \sigma_H^2)$ over action function parameters. Then, we have

$$\alpha_{H_{m_j}}^* = \frac{\sum_{k,i} p(\mu_{k,i} = m_j) \cdot \alpha_{H_{k,i}}}{\sum_{k,i} p(\mu_{k,i} = m_j)}$$

$$(\sigma_{H_{m_j}}^*)^2 = \frac{\sum_{k,i} p(\mu_{k,i} = m_j) \cdot (\alpha_{H_{k,i}} - \alpha_{H_{m_j}}^*)(\alpha_{H_{k,i}} - \alpha_{H_{m_j}}^*)^T}{\sum_{k,i} p(\mu_{k,i} = m_j)}$$

Solving for the parameters of $G_{m_{j_1}}^{m_{j_2}}$ is more challenging, since there can be multiple kinds of expressions in the grammar that are switching conditions, which correspond to discrete parameters, and we need to optimize over these discrete choices. To do so, we perform a greedy search over these discrete choices (see Section A.3 for details on the greedy strategy). For each choice considered during the greedy search, we encode Eq (12) as a numerical optimization and solve it to compute the corresponding means $\alpha_{G_{m_{j_1}}^{m_{j_2}}}^*$ and standard deviations $\sigma_{G_{m_{j_1}}^{m_{j_2}}}^*$ Then, we choose the discrete choice that achieves the best objective value according to Eq (12).

Computing the optimal parameters for switching conditions is more expensive than doing so for action functions; thus, on each student iteration, we iteratively solve Eq (10) and Eq (11) multiple times, but only solve Eq (12) once.

The EM method does not guarantee global optima but usually works well in practice. In addition, since computing the switching conditions is expensive, we had to restrict the number of EM iterations. However, note that even if the EM algorithm didn't converge, our overall algorithm can still recover by using additional teacher-student interactions.

The alternate method would be to run the EM algorithm multiple times/longer to get better results per student iteration, and "maybe" reduce the total number of teacher-student iterations. We say "maybe" because the EM algorithm might have already converged to the global optima, making the extra EM iterations useless. The trade-off between our approach and this alternative depends on whether the teacher's algorithm or the student's algorithm is expensive for a particular benchmark.

However, from Figure 15, we can see that some of our benchmarks already use very few ($< 5$) teacher-student iterations (Car, QuadPO, Pendulum, Mountain car, and Swimmer). Of the other three benchmarks that needed many iterations, for two of them (Cartpole and Acrobot), the student's algorithm is as expensive as the teacher's algorithm. This justifies our decision to not run the EM algorithm multiple times/longer.

## A.3 SYNTHESIZING SWITCHING CONDITIONS

Next, we describe how we search over the large number of discrete choices in the grammar for switching conditions. It is not hard to show that in Eq (12), the objectives for the switching condition parameters $G_{m_{j_1}}^{m_{j_2}}$ corresponding to different transitions $(m_{j_1}, m_{j_2})$ decompose into separate problems. Therefore, we can perform the search for each transition $(m_{j_1}, m_{j_2})$ separately. For each transition, the naïve approach would be to search over the possible derivations in the context-free grammar for switching conditions to some bounded depth. However, this search space is exponential in the depth due to the productions $B ::= B \wedge B$ and $B ::= B \vee B$. Thus, we employ a greedy search strategy to avoid the exponential blowup.

Intuitively, our search strategy is to represent switching conditions as a kind of decision tree, and then perform a greedy algorithm to search over decision tree[1]. Our search strategy is similar to (but simpler than) the one in Bielik et al. (2017). In particular, we can equivalently represent a switching condition as a decision tree, where the internal nodes have the form $\mathbf{o}[i] \leq \alpha$ or $\mathbf{o}[i] \geq \alpha$ (where $i \in \{1, ..., d_O\}$ and $\alpha \in \mathbb{R}$ are parameters), and the leaf nodes are labeled with "Switch" or "Don't switch"—e.g., Figure 7 shows two examples of switching conditions expressed as decision trees. Then, our algorithm initializes the switching condition to a single leaf node—i.e., $G_{\text{cur}} \leftarrow$ "Switch". At each step, we consider switching conditions $G \in \text{next}(G_{\text{cur}})$ that expand a single leaf node of $G_{\text{cur}}$; among these, we choose $G_{\text{cur}}$ to be the one that minimizes a loss $\text{cost}(G)$.

---

[1]However, our algorithm is not very similar to greedy decision tree learning algorithms.

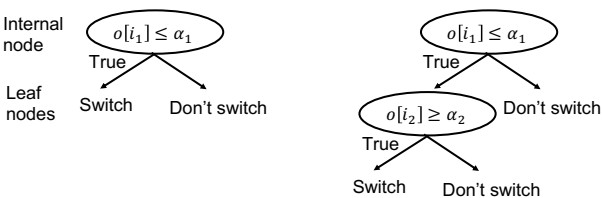

Figure 7: Switching conditions represented as decision trees.

---

**Algorithm 1** Greedy algorithm for learning switching conditions.

---
**procedure** LEARNSWITCHINGCONDITION
$\quad G_{\text{cur}} \leftarrow$ "Switch"
$\quad$ **while** $|G_{\text{cur}}| < N$ **do**
$\quad\quad G_{\text{cur}} \leftarrow \arg\min_{G \in \text{next}(G_{\text{cur}})} \text{cost}(G)$
$\quad$ **return** $G_{\text{cur}}$

---

More precisely, to construct $\text{next}(G_{\text{cur}})$, we iterate over all leaf nodes $L \in \text{leaves}(G_{\text{cur}})$, and all expressions $E \in \mathcal{E}$, where

$$\mathcal{E} = \left\{ \text{if } \mathbf{o}[i] \sim \alpha \text{ then "Switch" else "Don't Switch"} \ \middle| \ i \in \{1, ..., d_O\}, \ \alpha \in \mathbb{R}, \ \sim \in \{\geq, \leq\} \right\}$$

Here, $\sim \in \{\geq, \leq\}$ is a inequality relation, $i \in \{1, ..., d_O\}$ is a component of $\mathbf{o}$, and $\alpha \in \mathbb{R}$ is a threshold. For each pair $L$ and $E$, we consider the decision tree $G$ obtained by replacing $L$ with $E$ in $G_{\text{cur}}$. The set $\text{next}(G_{\text{cur}})$ contains all $G$ constructed in this way.

Next, the loss function $\text{cost}(G)$ is given by Eq (12). In each iteration, our algorithm optimizes $\text{cost}(G)$ over $G \in \text{next}(G_{\text{cur}})$, and updates $G_{\text{cur}} \leftarrow G$. To solve this optimization problem, we enumerate the possible choices $\sim$ and $i$ and use numerical optimization to compute $\alpha$ (since $\alpha$ is a continuous parameter). An example of a single iteration of our algorithm is shown in Figure 7. In particular, letting $G$ be the tree on the left and $G'$ be the tree on the right, the left-most leaf node of $G$ is expanded to get $G'$.

Our algorithm is summarized in Algorithm 1. Overall, our algorithm searches over $N \cdot (N-1) \cdot d_O$ different discrete structures, where $N$ is the number of nodes in the decision tree and $d_O$ is the length of the observation vector $\mathbf{o}$.

## B  IMPLEMENTATION DETAILS

### B.1  BENCHMARKS AND STATE-MACHINES STATISTICS

Figure 8 shows the statistics regarding the benchmarks such as the number of action variables and observation variables, and the set of initial states used for training and testing. Figure 8 also shows the different aspects of the grammar used to describe the space of possible state-machine policies. We are able to learn policies for these benchmarks using 2 to 4 distinct modes in the state-machine with either a constant or a proportional grammar for the action functions. We use the Boolean tree grammar of depth 1 or 2 for all the switching conditions.

For the Quad benchmark, the action variable is the acceleration of the quadcopter in the vertical direction. The observations include the position $x, y$, the velocities $v_x, v_y$, and the four sensors $ox_l, ox_u, oy_l, oy_u$ to describe the obstacle course in the near neighborhood. The QuadPO benchmark has the same action space as the Quad benchmark, but can only observe $x, y, v_x$, and $v_y$. The synthesized state-machine policies for these benchmarks are shown in Figure 16 and Figure 17. The action functions used for these benchmarks choose the acceleration to be proportional to $v_y$.

The goal for the Pendulum benchmark is to control the force (continuous) at the actuated link in order to invert the link. The observation space includes the full state, i.e., the angle $\theta$ and the angular velocity $\omega$. Figure 18 shows the synthesized state-machine policy for the pendulum benchmark.

| Bench | #A | #O | $X_0^{\text{train}}$ | $X_0^{\text{test}}$ | # modes | A_G | C_G |
|---|---|---|---|---|---|---|---|
| Car | 2 | 5 | d ~ [12,13.5]m | d ~ [11,12]m | 3 | Constant | Boolean tree (depth 1) |
| Quad | 1 | 8 | x dist = 40m | x dist = 80m | 2 | Proportional | Boolean tree (depth 1) |
| QuadPO | 1 | 4 | x dist = 60m | x dist = 120m | 2 | Proportional | Boolean tree (depth 1) |
| Pendulum | 1 | 2 | mass ~ [1,1.5]kg | mass ~ [1.5,5]kg | 2 | Constant | Boolean tree (depth 2) |
| Cartpole | 1 | 4 | time = 5s, len = 0.5 | time = 300s, len = 1.0 | 2 | Constant | Boolean tree (depth 2) |
| Acrobot | 1 | 4 | masses = [0.2,0.5] | masses = [0.5,2] | 2 | Constant | Boolean tree (depth 2) |
| Mountain car | 1 | 2 | power = [5,15]e-4 | power = [3,5]e-4 | 2 | Constant | Boolean tree (depth 1) |
| Swimmer | 3 | 10 | len = 1 unit | len = 0.75 unit | 4 | Proportional | Boolean tree (depth 2) |

Figure 8: Summary of our benchmarks. #A is the action dimension, #O is the observation dimension, $X_0^{\text{train}}$ is the set of initial states used for training, $X_0^{\text{test}}$ is the set of initial states used to test inductive generalization, # modes is the number of modes in the state machine policy, and A_G and C_G are the grammars for action functions and switching conditions, respectively. Depth of C_G indicates the number of levels in the Boolean tree.

The Cartpole benchmark consists of a pole attached to a cart. The goal is to keep the pole upright by applying a continuous force to move the cart to the right or to the left. The observations include the position $x$, the velocity $v$ of the cart, the angle $\theta$, and the angular velocity $\omega$ of the pole. The synthesized solution is shown in Figure 19.

The Acrobot benchmark is similar to the Pendulum benchmark but with two links; only the top link can be actuated, and the goal is to drive the bottom link above a certain height. The observations are the angles $\theta_1, \theta_2$ and the angular velocities $\omega_1, \omega_2$ of the two links. For this benchmark, we vary the mass of the links between the training and the test distributions. The synthesized solution is shown in Figure 20.

For the Mountain car benchmark, the goal is to drive a low powered car to the top of a hill. An agent has to drive back and forth to gain enough momentum to be able to cross the hill. The agent controls the force (continuous) to move the car to the right or left and observes the position $x$ and the velocity $v$ at every timestep. We vary the power of the car between the training and the test distributions. The synthesized solution is shown in Figure 21.

The Swimmer benchmark is based on the Mujoco's swimmer. To make this benchmark more challenging, we use 4 segments instead of 3. There are three actions that control the torques at the joints and the goal is to make the swimmer move forward through a viscous liquid. The agent can observe the swimmer's global angle $\theta$, the joint angles $(\theta_1, \theta_2, \theta_3)$, the swimmer's global angular velocity $\omega$, the angular velocities of the joints $(\omega_1, \omega_2, \omega_3)$, and the velocity of the center of mass $(v_x, v_y)$. We vary the length of the segments between the training and the test distributions. The actions are chosen to be proportional to their corresponding angles. The synthesized state machine policy is shown in Figure 22.

## B.2 HYPER-PARAMETERS

There are three main hyper-parameters in our algorithm:

- The maximum number of segments/modes in a loop-free policy. A large number of segments makes the teacher's numerical optimization slow, while a small number of segments might not be sufficient to get a high reward.

- The maximum time that a segment can be executed for in a loop-free policy. This maximum time constraint helps the numerical optimization to avoid local optima that arise from executing a particular (non-convex) action function for too long.

- The parameter $\lambda$ in Section 4.1. This parameter strikes a balance between preferring high-reward loop-free policies versus preferring policies that are similar to the state-machine learned so far.

The first two parameters solely affect the teacher's algorithm; thus, we choose them by randomly sampling from a set and select the one that produces high-reward loop-free policies. We use $\lambda = 100$ for all our experiments.

## B.3 THE DIRECT-OPT BASELINE

For this baseline, we convert the problem of synthesizing a state machine policy into a numerical optimization problem. To do this, we first encode the discreteness in the grammar for switching conditions into a continuous one-hot representation. For example, the set of expressions $\mathbf{o}[i] \leq \alpha_0$ or $\mathbf{o}[i] \geq \alpha_0$ are encoded as $\alpha_s(\alpha_1\mathbf{o}[1] + \alpha_2\mathbf{o}[2] + \cdots \alpha_n\mathbf{o}[n]) \leq \alpha_0$ with constraints $-1 \leq \alpha_s \leq 1$, $\forall i \in \{1, ..., n\}$. $0 \leq \alpha_i \leq 1$ and $\sum_{i=1}^{n} \alpha_i = 1$. The choices between the leaf expressions, conjunctions, and disjunctions are also encoded in a one-hot fashion. We also tried an encoding without the extra constraints on $\alpha$—i.e., the switching conditions are linear functions of the observations. We would expect the linear encoding to be less generalizable than the one-hot encoding. However, we found that it is hard to even synthesize a policy that works well on the training set with either of the encodings.

Another difficulty with direct optimization is that we need to optimize the combined reward from all the initial states at once. In contrast, the numerical optimization performed by the teacher in our approach can optimize the reward for each initial state separately. To deal with issue, we use a batch optimization technique that uses 10 initial states for every batch and seeds the starting point of the numerical optimization for each batch with the parameters found so far. It also restarts the process with a random starting point if the numerical optimization stalls. We carryout this process in parallel using 10 threads until either a solution is found or the time exceeds 2 hours.

## B.4 RL BASELINES

We use the PPO2 implementation from OpenAI Baselines (Dhariwal et al., 2017) with the standard MLP and LSTM networks for our RL baselines using $10^7$ timesteps for training.

**Environment featurization.** We used the same action spaces, observation spaces, and the set of initial states that we used for our approach. One exception is the Car benchmark, for which we appended the observation vector with observations from the previous timestep. This modification was essential for the RL baseline to achieve a good performance on the training dataset.

**Designing reward functions.** While our approach takes in a safe specification $\phi_S(\mathbf{x}) \leq 0$ and a goal specification $\phi_G(\mathbf{x}) \leq 0$, the RL baselines need a reward function. For the classic control problems such as cartpole, pendulum, acrobot, mountain car and swimmer, we used the standard reward functions as specified by their OpenAI environments. For Quad and QuadPO benchmarks, since the goal is to avoid collisions for as long as possible, we use a reward of 1 for every timestep that the agent is alive and the agent is terminated as soon as it collides with any of the obstacles. Designing the reward function for the Car benchmark was tricky, because this benchmark has both a goal and a safety specification, and finding a right balance between them is crucial for learning. We tried various forms of rewards functions and finally, found that the following version achieves better performance on the training distribution (on the metric that measures the fraction of roll-outs that satisfy both the goal and the safety property):

$$r(\mathbf{x}, \mathbf{a}) = -\phi_G(\mathbf{x})^+ + \begin{cases} -L & \text{if } \phi_S(\mathbf{x}) > 0 \\ 0 & \text{otherwise} \end{cases}$$

which adds the numerical error for not satisfying the goal with a constant negative error $(-L)$ if the safety specification is violated at any time-step. We tried different values for $L \in \{0.1, 1, 2, 10, 20\}$ and found that $L = 10$ achieves the best performance on the training distribution.

**Hyper-parameters search.** We performed a search over the various hyper-parameters in the PPO2 algorithm. We ran 10 instances of the PPO2 algorithm with parameters uniformly sampled from the space given below, and chose the one that performs well on the training distribution. This sampling is not exhaustive, but our results in Figure 4 (left most) show that we did find parameters that achieve good training performances for most of our benchmarks.

- The number of training minibatches per update, nminibatches $= \{1, 2, 4, 8, 16, 32, 64, 128, 256, 512, 1024, 2048\}$. For the lstm network, we set this hyper-parameter to 1.

|  |  | Performance on Train dist. | | Performance on Test dist. | |
|---|---|---|---|---|---|
| Bench | Algorithm | G | T_G | G | T_G |
| Acrobot | Ours | 0.08 | 7.9s | 0.02 | 31.8s |
|  | RL | 0.16 | 6.5s | 0.0 | 45.2s |
|  | Direct-opt | ⊥ | ⊥ | ⊥ | ⊥ |
| Mountain car | Ours | 0.001 | 168.5s | 0.008 | 290.1s |
|  | RL | 0.0 | 98.7s | 0.0 | 214.7s |
|  | Direct-opt | 0.006 | 105.3s | 2.18 | 216.0s |

Figure 9: Experiment results for additional benchmarks. G is the average goal error (closer to 0 is better). T_G is the average number of timesteps to reach the goal (lower the better). ⊥ indicates timeout. We can see that both our approach and RL generalizes for these benchmarks.

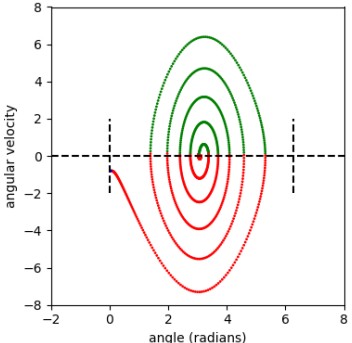 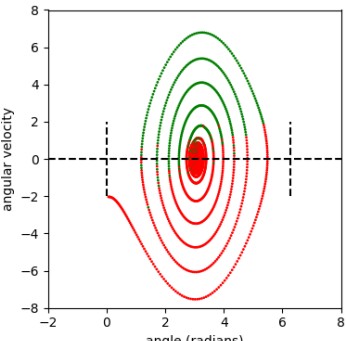

Figure 10: Trajectories taken by our state machine policy (left) and the RL policy (middle) on Pendulum for a test environment (i.e., heavier pendulum). Green (resp., red) indicates positive (resp., negative) torque. Our policy performs optimally by using positive torque when angular velocity $\geq 0$ and negative torque otherwise. In contrast, the RL policy performs sub-optimally (especially in the beginning of the trajectory).

- The policy entropy coefficient in the optimization objective, ent_coef = $\{0.0, 0.01, 0.05, 0.1\}$.

- The number of training epochs per update, noptepochs $\in \{3, ..., 36\}$.

- The clipping range, cliprange = $\{0.1, 0.2, 0.3\}$.

- The learning rate, lr $\in [5 \times 10^{-6}, 0.003]$.

## C  ADDITIONAL RESULTS

### C.1  ADDITIONAL PERFORMANCE RESULTS

Figure 9 shows the training and test performance for the acrobot and mountain car benchmarks. We can see that both our approach and RL generalizes for these benchmarks.

Figure 10 qualitatively analyzes the policies learned by our approach versus RL for the Pendulum benchmark. We can see that the RL policy performs slightly sub-optimally compared to our policy.

Figure 11 shows the trajectories from the learned state machine policy and RL policy on Swimmer for a train environment and a test environment. While both policies generalize, the swimmer with the state machine policy is slightly faster (it takes about 35s to cover a distance of 10 units while the RL policy takes about 45s).

Figures 12, 13, & 14 show the action versus time plots for the various benchmarks using the learned state-machine policies and neural network policies. We can see that state-machine polices produce smooth actions, whereas the RL policies do not.

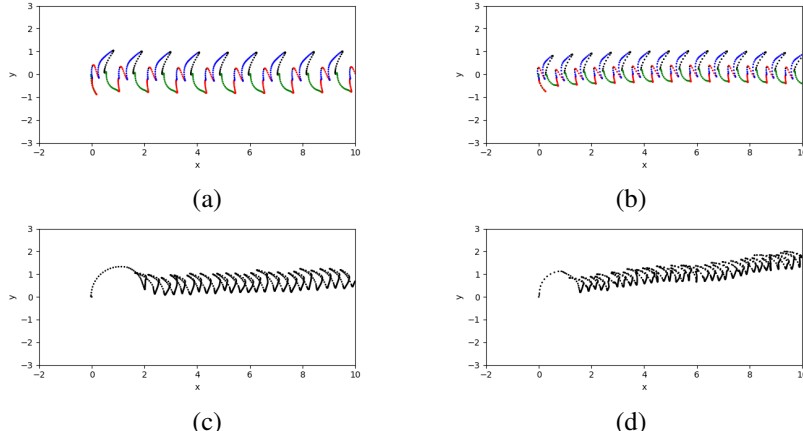

Figure 11: Trajectories taken by our state machine policy on Swimmer for (a) a train environment with segments of length 1, and (b) a test environment with segments of length 0.75. The colors indicate different modes. The axes are the $x$ and $y$ coordinates of the center of mass of the swimmer. Trajectories taken by the RL policy on Swimmer for (c) a train environment, and (d) a test environment. While both policies generalize, the swimmer with the state machine policy is slightly faster (it takes about 35s to cover a distance of 10 units while the RL policy takes about 45s).

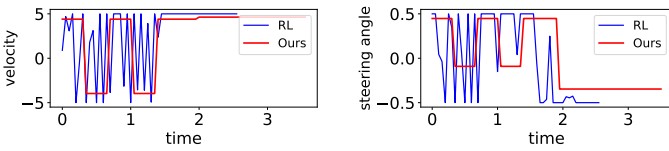

Figure 12: Action vs time graphs for the car benchmark for both our policy (red) and the neural network policy (blue). Left shows the velocity of the agent and Right shows the steering angle.

## C.2 ANALYSIS OF RUNNING TIME

Figure 15 shows the synthesis times for various benchmarks. It also shows the number of student-teacher iterations, and the time spent by the teacher and the student separately. The teacher optimizes the loop-free policies for different initial states in parallel. The student optimizes the switching conditions between different pairs of modes in parallel. We used a parallelized implementation with 10 threads, and report the wall clock time.

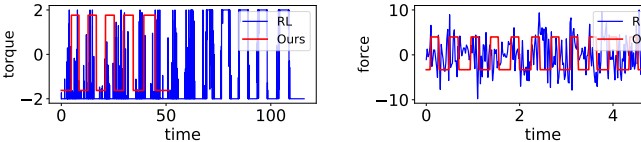

Figure 13: Action vs time graphs for the pendulum benchmark (left) and the cartpole benchmark (right) for both our policy (red) and the neural network policy (blue).

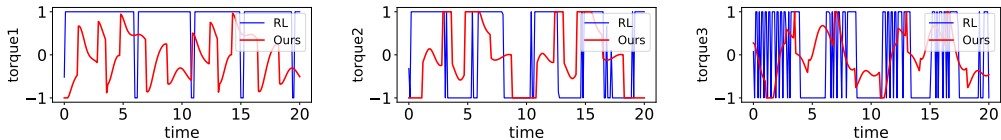

Figure 14: Action vs time graphs for the swimmer benchmark for the three torques at the three different joints of the swimmer. Blue line is for the neural network policy and red line is for the state machine policy.

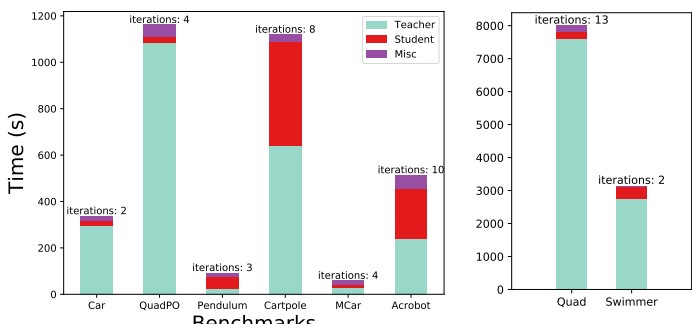

Figure 15: Synthesis times (in seconds, wall clock time) for learning state machines policies for the different benchmarks. The plot breaks down the total synthesis time into time taken by the teacher, the student and other miscellaneous parts of the algorithm. Misc. mostly includes the time spent for checking convergence at every iteration. The plot also shows the number of teacher-student iterations taken for each benchmark.

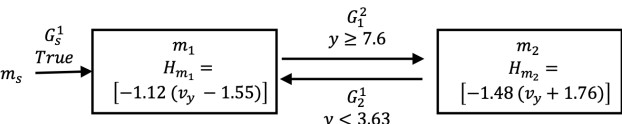

Figure 16: Synthesized state-machine policy for the Quad benchmark.

Figure 17: Synthesized state-machine policy for the QuadPO benchmark.

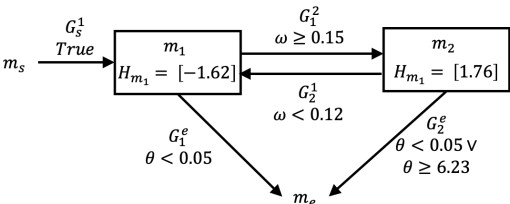

Figure 18: Synthesized state-machine policy for Pendulum.

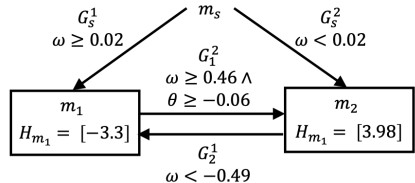

Figure 19: Synthesized state-machine policy for Cartpole.

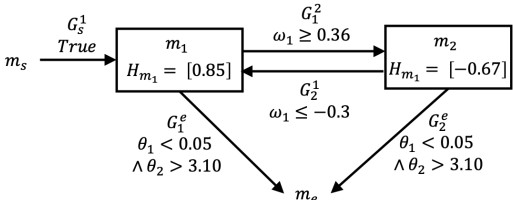

Figure 20: Synthesized state-machine policy for Acrobot.

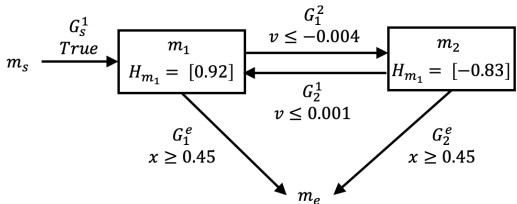

Figure 21: Synthesized state-machine policy for Mountain car.

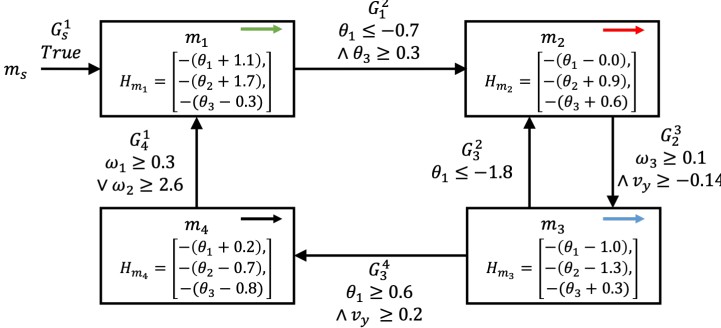

Figure 22: Synthesized state-machine policy for Swimmer.

