# OpenReview forum: "Synthesizing Programmatic Policies that Inductively Generalize"
_ICLR.cc/2020/Conference — Accept (Poster)_

### Official Review · AnonReviewer2 · 2019-10-24
**Official Blind Review #2**

**Rating:** 6

**Review:**

This work proposes a framework for structuring policies using finite state machines, training a teacher-student setup using variational inference. The method is tested on a suite of standard control problems against PPO with / without memory and against simple numerical optimisation, analysing both performance and some degree of generalisation.


Overall, I like both the problem setting (constraining / structuring policies using SMs), and the proposed modeling and optimisation. In particular, the teacher-student setup makes sense to me in the way it has been casted under VI, and I would like to see it explored further.

I have however a few major issues that prevent me from recommending acceptance of the work:

1. The experiment section is generally lacking in terms of implementation details. The authors don't specify anything about models structure or details about the employed state machines, and do not seem to have included details about their direct optimisation baseline, environment featurisation, hyperparameters used across their experiments, and so on. I very much doubt I would be able to reproduce the results based only on the manuscript.

2. The quality of the policies depend heavily on how close the proposed state machine matches the underlying dynamics of each task. Since - very often - complex tasks are hard to optimally describe without producing very large state machines, I would have liked to see the method tested against poor and/or adversarial specifications, to see whether empirically how well the student-teacher optimisation system can recover under such constraints.

3. Casting the problem as a POMDP, while technically fine (and in most cases reasonable), doesn't seem to provide any significant advantage to the method, and seems to only be adding noise in the formalisms described across the paper. Since the method introduces notation that a lot of RL researchers are not necessarily familiar with, I would suggest the author to try to simplify it where possible. [Also please note that I haven't re-derived section 4.1 under this assumption, so correct me if I'm wrong on this.]


At this point, I am recommending a weak rejection, however I will be happy to significantly reconsider my overall assessment provided that at least point (1) is decently addressed (and ideally I get some response wrt. points 2 and 3).


**Experience Assessment:**

I have published in this field for several years.

**Review Assessment: Checking Correctness Of Derivations And Theory:**

I assessed the sensibility of the derivations and theory.

**Review Assessment: Checking Correctness Of Experiments:**

I carefully checked the experiments.

**Review Assessment: Thoroughness In Paper Reading:**

I read the paper thoroughly.

---

> ### Author Response · Authors · 2019-11-13
> **Response to Review #2**
>
> Thank you for thoroughly reading the paper and your valuable comments. We address your concerns below:
>
> **** Implementation details and direct opt baseline ****
>
> We added details about the environments, state machines, hyper-parameters and the direct-opt baseline in Appendix B.3 in our paper. We also show all synthesized state-machines in Figures 14 to 20.
>
> **** Scaling to synthesizing large state machines ****
>
> We have added a new, more complex benchmark -- namely, the MuJoCo swimmer (extended to 4 segments instead of 3 to make the task more challenging) -- to our paper. This benchmark has control inputs R^3 and observation space R^10. The state-machine policy synthesized using our algorithm has 4 different modes. (Section 5, Figure 10, Figure 20)
>
> Overall, the focus of our paper is on addressing problems where a relatively simple behavior must be repeated a certain number of times to solve the given task. We believe these problems are pervasive -- for example, many motor tasks such as walking, running, jumping, swimming, etc. all rely on this kind of a policy. Yet, neural network policies have difficulty solving these tasks in a generalizable way. The key premise behind our approach is that compact state-machine policies can represent policies that both have good performance and are generalizable for this class of problems. Indeed, our algorithm solves all of our benchmarks using state-machine policies with at most 4 modes and switching conditions of depth at most 2.
>
> For example, consider the autonomous car example in Figure 1. The task in Figure 1d (i.e. when the gap between cars is very small) is significantly complex that the neural network baseline was not able to solve (even when trained directly on those initial states; see Figure 3 rightmost). However, a small state-machine policy with only 3 modes was able to solve this task.
>
> We certainly agree with the reviewer that it will be interesting to see how the algorithm scales for larger state machines. However, this problem is qualitatively different from the one we are solving, and different algorithms would be needed to solve it. Overall, we believe that state-machines are most useful when only a few states are required. When a large number of states are needed, then the number of possible transition structures grows exponentially, making it unlikely that we can learn the “true” structure without having an exponential amount of both training data and computation time. For these cases, we believe recurrent neural network policies such as LSTMs may remain the better choice.
>
> **** Casting the problem as a POMDP ****
>
> We use a POMDP formulation because our policies have internal state and some of our benchmarks (e.g., QuadPO) have different observation vectors and the state vectors. We thought about focusing on MDPs to simplify notation, but concluded that it would not significantly help. The aspect of our approach that complicates our formalization is that our policies keep internal state. However, this aspect is central to our approach, since we are both positing that internal state is necessary, and that it is hard to learn programmatic policies with internal state.

---

> > ### Comment · AnonReviewer2 · 2019-11-14
> > **Response to rebuttal**
> >
> > >We added details about the environments, state machines, hyper-parameters and the direct-opt baseline in Appendix B.3 in our paper. We also show all synthesized state-machines in Figures 14 to 20.
> >
> > Thank you for adding these, I'm satisfied with the info given by appendices A.3 - B.3 wrt your own method and the direct-op baselines, however would it be possible to also add details about your RL baselines? The strength of the experimental section relies a lot on the comparison against these, so readers (and I) would most likely appreciate some more details about them.
> >
> >
> > Regarding point (2), I think you make a fair rebuttal, however I would like the narrative behind your response to be clearly expressed in the manuscript. In particular, Section 1 remains quite vague and too general to provide reasonable expectations to the reader, so I would suggest to put some time into improving it.
> >
> >
> > That said, I am happy for the moment to change my score to a weak accept, considering the amount of good additional work that was put into the manuscript.

---

> > > ### Author Response · Authors · 2019-11-14
> > > **Response to additional comments**
> > >
> > > Thank you for going through our rebuttal and updating your score.
> > >
> > > **** RL baseline details ****
> > > We added details about the RL baselines in appendix B.4. To summarize, we used the PPO2 algorithm from OpenAI baselines [1] and tried various hyper-parameters and reward functions (for the new benchmarks such as Car in figure 1), and chose one that performs well on the training distribution.
> > >
> > > [1] Prafulla Dhariwal, Christopher Hesse, Oleg Klimov, Alex Nichol, Matthias Plappert, Alec Radford, John Schulman, Szymon Sidor, Yuhuai Wu, and Peter Zhokhov.   Openai baselines. https://github.com/openai/baselines, 2017.
> > >
> > >
> > > **** Update introduction to reflect the rebuttal ****
> > > We updated the introduction to set reasonable expectations for a reader.

---

### Official Review · AnonReviewer1 · 2019-10-25
**Official Blind Review #1**

**Rating:** 8

**Review:**

This paper can be viewed as being related to two bodies of work:

(A) The first is training programmatic policies (e.g., https://arxiv.org/abs/1907.05431).  The most popular idea is to use program synthesis & imitation learning to distill from a programmatic policy from some oracle policy.

(B) The second is training compact policies using a complex model-based controller (e.g., Guided Policy Search).   The idea is to use a step-wise model-based controller to design a good trajectory that maximizes reward, while not deviating too far from the current policy.  Then the new policy is learned from this trajectory.

The authors contrast with (A) via "our student does not learn based on examples provided by the teacher, but is trained to mimic the internal structure of the teacher". The authors contrast with (B) in part by claiming that "the teacher must mirror the structure of the student", which is supposedly harder.

Thus, it seems much of the intellectual merit & novelty lies how the proposed method tackles this "structure" problem, from both the teacher and the student side.  However, I'm having a hard time appreciating this aspect of the proposed approach.  I'm also confused by how the "student does not learn based on examples provided by the teacher" if it's doing imitation learning on trajectory-level demonstrations. Can the authors elaborate on this point further?

The experiments seem ok.  The idea of training programmatic polices that "inductively generalize" has been done before on arguably more difficult tasks (see Table 2 in https://arxiv.org/abs/1907.05431).   To contrast with this prior result, it seems the main point is that prior work relied on domain-specific program synthesizers.  Can the authors elaborate on this point further?

Minor comments:

-- Adaptive teaching is a pretty ambiguous term, and I think misleading within ML community (cf. https://arxiv.org/abs/1802.05190). I recommend a different algorithm name.

-- Deriving the variational objective is a lot of work to reduce it to just trajectory design.  Seems overkill.



------------------------------------------------------------
Updates after Author Response
------------------------------------------------------------

I increased my score to accept.  I think this is a worthy contribution, and the authors did an excellent job addressing my concerns.

**Experience Assessment:**

I have published one or two papers in this area.

**Review Assessment: Checking Correctness Of Derivations And Theory:**

I carefully checked the derivations and theory.

**Review Assessment: Checking Correctness Of Experiments:**

I carefully checked the experiments.

**Review Assessment: Thoroughness In Paper Reading:**

I read the paper thoroughly.

---

> ### Author Response · Authors · 2019-11-13
> **Response to Review #1**
>
> Thank you for thoroughly reading the paper and your valuable comments. We address your concerns below:
>
> **** Importance of the state-machine structure and difference with prior work ****
>
> Prior work in [1] and [2] has shown that programmatic policies achieve better interpretability and generalization over neural network policies. For generalizability, they have shown that a policy learned for one particular track of the Torcs game can be easily transferred to a couple of other tracks. In contrast, we focus on a different kind of generalization, namely, to performing a repetitive behavior arbitrarily many times. We train on problem instances that only require a small number of repetitions of a behavior (e.g., 1 to 3), but the policies we learn generalize to problem instances where the behavior must be repeated an arbitrarily large number of times.
>
> Prior work uses stateless programs, which makes learning generalizable repetitive behaviors hard, e.g., a stateless program cannot internally keep track of the progress made within each repetition and decide the next action based on that progress. In contrast, we use state-machine based policies that have internal state. With state-machine policies, the repetitive behavior in the tasks can be explicitly exposed (for ex, a set of modes that are connected in a cyclic order represents a loop), and the internal state (i.e. which mode the SM is in) can be used to keep track of the progress within a loop iteration.
>
> Synthesizing state-machine policies is substantially more challenging than previous work, since it involves latent states that are unobserved. A typical approach is to use a teacher such as a DNN to guide the search [1,2]; however, we found that DNNs are unable to learn the “correct” latent states (i.e., the ones they learn do not generalize).
>
> Our teacher is designed to avoid these problems. First, our teacher also has an explicit notion of modes, and hence, it can discover the “correct” modes that solve the task in a generalizable way, which the student will imitate. Second, our representations also make it easy for the student to enforce the repetitive structure in the modes of the loop-free policies learned by the teacher.
>
> In principle, the internal state can be replaced with the history of agent’s states. However, for some environments, a long history might be needed to replicate the internal state. Using such a long history would dramatically increase the dimension of the state space, making it very hard to generalize robustly.
>
>
> **** Examples vs trajectory traces ****
>
> We apologize for the confusion. When we said that “the prior works assume a domain-specific program synthesizer that can learn programmatic policies given a supervised dataset”, we mean that they can learn a programmatic policy based on “state-action pairs” sampled from the neural network.
>
> On the other hand, the student in our algorithm needs to learn the state-machine policy from entire “trajectory traces” to learn the internal state.  In particular, each trajectory trace consists of the sequence of states and actions from the initial state to the goal state visited by the teacher, but also encodes which states correspond to mode changes for the (loop-free) teacher. In the teacher’s iteration, the teacher mode changes are regularized to align more closely with the possible student mode changes. As a consequence, in the student’s iteration, it is easier for the student to mimic the teacher’s mode changes.
>
> Leveraging this connection between the teacher structure and student structure is critical for us to be able to learn state-machine policies. In contrast, existing approaches treat the teacher as a blackbox, and learn purely from the state-action examples obtained from the teacher. They can do so since there is no latent information that the student needs to learn. In contrast, in our setting, it is very hard to learn the latent mode transitions without supervision. Using a teacher with mode changes alleviates this challenge.
>
>
> [1] Abhinav Verma, Vijayaraghavan Murali, Rishabh Singh, Pushmeet Kohli, and Swarat Chaudhuri. Programmatically interpretable reinforcement learning. arXiv preprint arXiv:1804.02477, 2018.
> [2] Abhinav Verma, Hoang Minh Le, Yisong Yue, and Swarat Chaudhuri.  Imitation-projected policy gradient for programmatic reinforcement learning. CoRR, abs/1907.05431, 2019.  URL
> http: //arxiv.org/abs/1907.05431.

---

### Official Review · AnonReviewer3 · 2019-10-31
**Official Blind Review #3**

**Rating:** 6

**Review:**

# Summary

This paper proposes a technique for synthesis of state machine policies for a simple continuous agent, with a goal of
them being generalizable to out-of-distribution test conditions. The agent is modeled as a state machine with constant
or proportional actions applied in each node (regime), and switching triggers between the regimes represented as
length-2 boolean conditions on the observations. The technique is evaluated on 7 classic control environments, and found
to outperform pure-RL baselines under "test" conditions in most of them.

# Review

I am not an expert in RL-based control, although I'm familiar with the recent literature that applies formal methods to
these domains. I find the studied settings valuable albeit fairly limited, but the paper's method undeniably shows
noticeable improvement on these settings. Inductive generalization is an important problem, and the authors' approach of
limiting the agent structure to a particular class of state-machine policies is a reasonable solution strategy.

That said, the complexity of synthesizing a state machine policy clearly caused the authors to limit their supported
action and condition spaces considerably (Figure 6). That, I'm assuming, limits the set of applicable control
environments where optimization is still feasible. The authors don't provide any analysis of complexity or empirical
runtime of the optimization process. Breaking it down for each benchmark would allow me to appreciate the optimization
framework in Section 4 much more. As it stands, Section 4 describes a complex optimization process with many moving
parts, some of which are approximated (q* and p(τ|π,x₀)) or computed via EM iteration until convergence (π*). It is hard
to appreciate all this complexity without knowing where the challenges manifest on specific examples.

Section 4.2 needs an example, to link it to the introductory example in Figure 1. The "loop-free" policies of the
teacher are, in programmatic terms, _traces_ of the desired state machine execution (if I understand correctly), but
this is not obvious from just the formal definition.

The EM optimization for the student policy makes significant assumptions on the action/condition grammars. Namely, the
algorithm iterates over every possible discrete "sketch" of every program, and then numerically optimizes its continuous
parameters (Appendix A). When the action/condition grammars grow, the number of possible programs there increases
combinatorially. Is there a way to adapt the optimization process to handle more complex grammars, possibly with
decomposition of the problem following the program structure?

Section 5 needs a bit more details on the Direct-Opt baseline. It's unclear how the whole state machine policy (which
includes both discrete and continuous parts) is learned end-to-end via numerical optimization. Granted, the baseline
performs terribly, but would be great to understand how it models the learning problem in order to appreciate why it's
terrible.

Why were the "Acrobot" and "Mountain car" benchmarks removed from the main presentation of results?

**Experience Assessment:**

I have read many papers in this area.

**Review Assessment: Checking Correctness Of Derivations And Theory:**

I assessed the sensibility of the derivations and theory.

**Review Assessment: Checking Correctness Of Experiments:**

I carefully checked the experiments.

**Review Assessment: Thoroughness In Paper Reading:**

I read the paper thoroughly.

---

> ### Author Response · Authors · 2019-11-13
> **Response to Review #3 (1/2)**
>
> Thank you for thoroughly reading the paper and your valuable comments. We address your concerns below:
>
> **** Complexity of the algorithm limits the set of applicable environments ****
>
> We have added a new, more complex benchmark -- namely, the MuJoCo swimmer (extended to 4 segments instead of 3 to make the task more challenging) -- to our paper. This benchmark has control inputs R^3 and observation space R^10. The state-machine policy synthesized using our algorithm has 4 different modes. (Section 5, Figure 10, Figure 20)
>
> Overall, the focus of our paper is on addressing problems where a relatively simple behavior must be repeated a certain number of times to solve the given task. We believe these problems are pervasive -- for example, many motor tasks such as walking, running, jumping, swimming, etc. all rely on this kind of a policy. Yet, neural network policies have difficulty solving these tasks in a generalizable way. The key premise behind our approach is that compact state-machine policies can represent policies that both have good performance and are generalizable for this class of problems. Indeed, our algorithm solves all of our benchmarks using state-machine policies with at most 4 modes and switching conditions of depth at most 2.
>
> For example, consider the autonomous car example in Figure 1. The task in Figure 1d (i.e. when the gap between cars is very small) is significantly complex that the neural network baseline was not able to solve (even when trained directly on those initial states; see Figure 3 rightmost). However, a small state-machine policy with only 3 modes was able to solve this task.
>
> We certainly agree with the reviewer that it will be interesting to see how the algorithm scales for larger state machines. However, this problem is qualitatively different from the one we are solving, and different algorithms would be needed to solve it. Overall, we believe that state-machines are most useful when only a few states are required. When a large number of states are needed, then the number of possible transition structures grows exponentially, making it unlikely that we can learn the “true” structure without having an exponential amount of both training data and computation time. For these cases, we believe recurrent neural network policies such as LSTMs may remain the better choice.
>
> **** Empirical runtime analysis ****
>
> We add runtime analysis in Appendix C.2 and Figure 11 in our paper. We show the synthesis times for various benchmarks, the number of student-teacher iterations, and the time spent by the teacher and the student separately. We hope this will give the reviewer a better perspective on the different parts of the algorithm
>
> **** Scaling the algorithm with the complexity of the grammar ****
>
> For the action functions, the user decides whether the action functions will be from a constant grammar or a proportional grammar. Furthermore, for proportional grammar, the user specifies which observation the action should be proportional to. There are no discrete choices in either of these two grammars. We believe that this is usually not difficult for the user to choose the best grammar to use; for instance, a user can easily say that the control for y-acceleration will be proportional be y-velocity and not x-velocity. An alternative would be to run the teacher’s algorithm with many different action function grammars and choose the grammar that resulted in high-reward loop-free policies.
>
> For the switching conditions, the discrete choices in the grammar will be learned by the synthesis algorithm along with the continuous parameters. We agree with the reviewer that the discrete enumeration involved here will quickly blow up with the size of the boolean expression. For this reason, we have a greedy algorithm that scales quadratically with the size of the expression rather than exponentially. We failed to mention this in the original paper, but updated it now (see Appendix A.3 in our paper for more details). All the experiments use this greedy algorithm for learning the switching conditions.
>
> **** Examples of loop-free policy ****
>
> We added two examples of loop-free policies in Figure 13 in the context of the example in Figure 1. We also show visualizations of trajectories learned by the teacher and the student for two different iterations of the algorithm in Figure 12.

---

> > ### Author Response · Authors · 2019-11-13
> > **Response to Review #3 (2/2)**
> >
> >
> > **** Details on the direct-opt baseline ****
> >
> > We added this to Appendix B.3 of our paper. To summarize, we use two different encodings to deal with the discreteness in the grammar for the switching conditions. First, we used a one-hot continuous encoding for the discrete options, and second, we used conditions that are linear functions of the observation vector. Both the encodings failed to achieve good performance even on the training distribution. We believe this is because of the discontinuities caused by the switching conditions (i.e., a small change in the parameters of the switching conditions) would lead to a big change in the trajectory, and hence, the reward.

---

> > > ### Comment · AnonReviewer3 · 2019-11-15
> > > **Thank you**
> > >
> > > Thank you for the detailed response, new experiments, and an improved revision. I read through the new PDF, and will keep my score at "Weak Accept".
> > > A few writing/motivational/structure recommendations:
> > >  •  Properly scoping the applicable tasks to those that can be well-described by state machines (as mentioned in your response) should be done early on, in the Introduction.
> > >  •  Examples of loop-free trajectories would be appreciated right after they are introduced, i.e. in Section 4.2.
> > >  •  The presented breakdown of run time is a good first step toward understanding challenges of the pipeline, but does not yet satisfactorily answer the question "Why a particular design choice – be it sampling, a particular approximation, EM, etc. – was made?" I suppose what I am looking for is some high-level insights for the reader, expressed in appropriate sections. Something like:
> > >
> > > "We handle $p(\tau|\pi,x_0)$ by (i) sampling a set of good trajectories $\tau^1, \dots, \tau^L$ from $x_0$, and (ii) approximating $p(\tau|\pi,x_0)$ with a weighted average over the top $\rho$ samples. This approximation is needed because all the trajectories are infeasible to compute. It seems to be appropriate in situations like our benchmarks X, Y, Z, and less appropriate in benchmarks P, Q, R. In the latter cases, the lower quality of the approximation presented by the teacher causes the student to require more EM steps until convergence."

---

> > > > ### Author Response · Authors · 2019-11-15
> > > > **Response to additional comments**
> > > >
> > > > Thank you for reading the rebuttal. We like your further recommendations and incorporated them into the paper.
> > > >
> > > > We updated the introduction to scope out the problem more concretely (and match our response above).
> > > >
> > > > We agree that the examples of loop-free trajectories should be in Section 4.2.
> > > >
> > > > We also agree that the high-level insights for the design decisions should be explicitly mentioned. Here, we point out some of our design decisions and reasons.  In general, most of the approximations are made for scalability reasons and we tried to use the best alternative available in the literature to handle similar issues.
> > > >
> > > > 1. Approximating $p(\tau | \pi, x_0)$
> > > >
> > > > Note that this probability will be computed as part of an objective for numerical optimization. Hence, it is very important to have a smooth closed-form representation of the probability that can be computed very efficiently (along with gradients).
> > > >
> > > > Symbolically computing the “true” probability is hard because of the discrete-continuous structure of $\pi$. Another alternative is to precompute the probabilities of all the trajectories ($\tau$) that can be derived from $\pi$. However, this is also infeasible because the number of trajectories is unbounded.
> > > >
> > > > Hence, we use an approach similar to the cross-entropy method [1], which is a standard tool in Monte Carlo estimation, and combinatorial and continuous optimization.
> > > >
> > > > For our benchmarks, we didn’t notice any improvement in the number of student-teacher iterations by increasing the number of samples ($\rho$) from the current value of 10. So, we believe we are not losing much information from this approximation.
> > > >
> > > > [1] Shie Mannor, Reuven Y Rubinstein, and Yohai Gat. The cross entropy method for fast policy search. In Proceedings of the 20th International Conference on Machine Learning (ICML-03), pp. 512–519, 2003.
> > > >
> > > >
> > > > 2. EM algorithm
> > > >
> > > > As shown in appendix A.1, computing the maximum likelihood estimate of the state machine ($\pi$) involves finding the latent mode-assignment variables ($\mu$) in addition to the parameters of the action functions and the switching conditions. Numerically optimizing this maximum likelihood objective is hard because it requires integrating over all possible choices for the latent variables. For example, if the teacher generates 10 loop-free policies every iteration and there are 10 modes in each loop-free policy, and 4 modes in the state-machine, the number of choices for the latent variables is 4^100, which makes the enumeration infeasible.
> > > >
> > > > The expectation-maximization method provides an efficient way for computing the maximum likelihood, by alternatingly optimizing for the latent variables and the state-machine parameters. This method does not guarantee global optima but usually works well in practice. In addition, since computing the switching conditions is expensive, we had to restrict the number of EM iterations.  However, note that even if the EM algorithm didn't converge, our overall algorithm can still recover by using additional teacher-student interactions.
> > > >
> > > > The alternate method would be to run the EM algorithm multiple times/longer to get better results per student iteration, and “maybe” reduce the total number of teacher-student iterations.  We say  “maybe” because the EM algorithm might have already converged to the global optima, making the extra EM iterations useless. The tradeoff between our approach and this alternative depends on whether the teacher’s algorithm or the student’s algorithm is expensive for a particular benchmark.
> > > >
> > > > However, from Figure 11, we can see that some of our benchmarks already use very few (< 5) teacher-student iterations  (Car, QuadPO, Pendulum, Mountain car, and Swimmer). Of the other three benchmarks that needed many iterations, for two of them  (Cartpole and Acrobot), the student’s algorithm is as expensive as the teacher’s algorithm. This justifies our decision to not run the EM algorithm multiple times/longer.

---

### Author Response · Authors · 2019-11-13
**Summary of changes to the paper**

We thank the reviewers for their feedback and valuable time. Based on the reviews, we updated the paper as follows:

1. Added a detailed algorithm for greedily learning switching conditions for large grammars (Appendix A.3)
2. Added implementation details (Appendix B), including a description of the direct-opt baseline (Appendix B.3)
3. Added an empirical evaluation of the running time of our algorithm (Appendix C.2)
4. Added visualizations of loop-free policies and student-teacher interactions (Appendix C.3, Figures 12 and 13)
5. Added synthesized policies for other benchmarks (Figures 14 to 20)
6. New complex example based on MuJoCo swimmer (Section 5, Figure 10, Figure 20)

---

> ### Author Response · Authors · 2019-11-14
> **Another update to the paper**
>
> We added details about the RL baselines (Appendix B.4).

---

> > ### Author Response · Authors · 2019-11-15
> > **More updates**
> >
> > Based on the new comments from the reviewers, we made the following changes:
> >
> > 1. Updated the introduction (Section 1) to make it clear that our approach does not focus on problems requiring larger state-machines.
> >
> > 2. Added examples of loop-free policies to Section 4.2.
> >
> > 3. Added intuitions for design choices in Section 4.2, 4.3 and Append A.2.

---

### Decision · Program_Chairs · 2019-12-19

**Decision:**

Accept (Poster)

**Comment:**

The authors consider control tasks that require "inductive generalization", ie
the ability to repeat certain primitive behaviors.
They propose state-machine machine policies, which switch between low-level
policies based on learned transition criteria.
The approach is tested on multiple continuous control environments and compared to
RL baselines as well as an ablation.

The reviewers appreciated the general idea of the paper.
During the rebuttal, the authors addressed most of the issues raised in the
reviews and hence reviewers increased their score.

The paper is marginally above acceptance.
On the positive side: Learning structured policies is clearly desirable but
difficult and the paper proposes an interesting set of ideas to tackle this
challenge.
My main concern about this work is:
The approach uses the true environment simulator, as the
training relies on gradients of the reward function.
This makes the tasks into planning and not an RL problems; this needs to be
highlighted, as it severly limits its applicability of the proposed approach.
Furthermore, this also means that the comparison to the model-free PPO baselines
is less meaningful.
The authors should clear mention this.
Overall however, I think there are enough good ideas presented here to warrant
acceptance.